# Vivid-ZOO: Multi-View Video Generation with Diffusion Model

**Bing Li**[*][†]    **Cheng Zheng**[*]    **Wenxuan Zhu**[*]    **Jinjie Mai**    **Biao Zhang**
**Peter Wonka**    **Bernard Ghanem**

King Abdullah University of Science and Technology

https://hi-zhengcheng.github.io/vividzoo/

## Abstract

While diffusion models have shown impressive performance in 2D image/video generation, diffusion-based Text-to-Multi-view-Video (T2MVid) generation remains underexplored. The new challenges posed by T2MVid generation lie in the lack of massive captioned multi-view videos and the complexity of modeling such multi-dimensional distribution. To this end, we propose a novel diffusion-based pipeline that generates high-quality multi-view videos centered around a dynamic 3D object from text. Specifically, we factor the T2MVid problem into viewpoint-space and time components. Such factorization allows us to combine and reuse layers of advanced pre-trained multi-view image and 2D video diffusion models to ensure multi-view consistency as well as temporal coherence for the generated multi-view videos, largely reducing the training cost. We further introduce alignment modules to align the latent spaces of layers from the pre-trained multi-view and the 2D video diffusion models, addressing the reused layers' incompatibility that arises from the domain gap between 2D and multi-view data. In support of this and future research, we further contribute a captioned multi-view video dataset. Experimental results demonstrate that our method generates high-quality multi-view videos, exhibiting vivid motions, temporal coherence, and multi-view consistency, given a variety of text prompts.

## 1 Introduction

Multi-view videos capture a scene/object from multiple cameras with different poses simultaneously, which are critical for numerous downstream applications [5, 55, 62, 39, 40] such as AR/VR, 3D/4D modeling, media production, and interactive entertainment. More importantly, the availability of such data holds substantial promise for facilitating progress in research areas such as 4D reconstruction [44, 48], 4D generation [3, 49], and long video generation [9, 101] with 3D consistency. However, collecting multi-view videos often requires sophisticated setups [1] to synchronize and calibrate multiple cameras, resulting in a significant absence of datasets and generative techniques for multi-view videos.

In the meantime, diffusion models have shown great success in 2D image/video generation. For example, 2D video diffusion models [6, 23, 28, 81] generate high-quality 2D videos by extending image diffusion models [74, 83]. Differently, multi-view image diffusion models [80, 34, 53, 93] are proposed to generate multi-view images of 3D objects, which have demonstrated significant impact in 3D object generation [45], 3D reconstruction [66], and related fields. However, to the best of our knowledge, no other works have explored Text-to-Multi-view-Video (T2MVid) diffusion

---

[*]Equal contributions. [†] Corresponding author.

38th Conference on Neural Information Processing Systems (NeurIPS 2024).

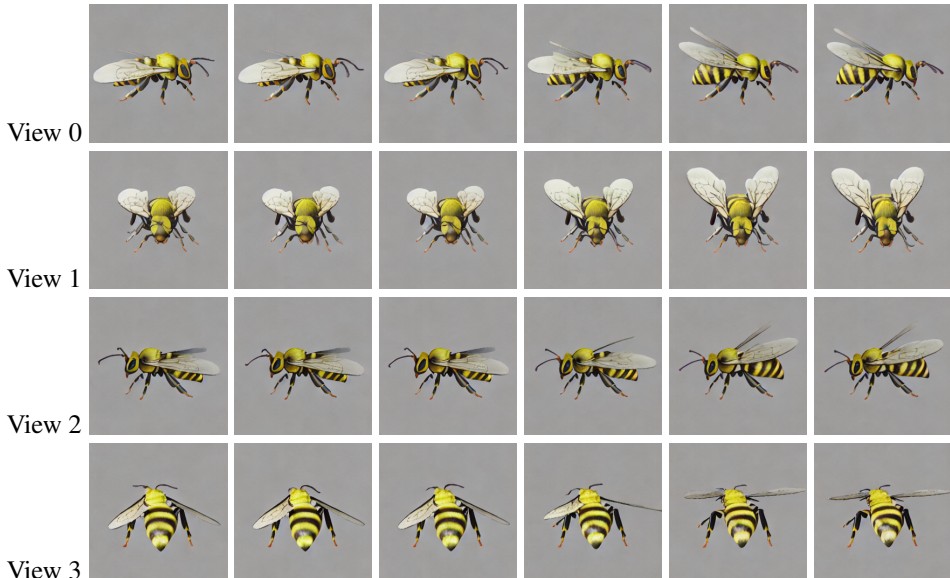

View 0

View 1

View 2

View 3

*Text prompt: a yellow and black striped wasp bee, 3d asset*

Figure 1: The proposed Vivid-ZOO generates high-quality multi-view videos of a dynamic 3D object from text. Each row illustrates six frames drawn from a generated video for one viewpoint.

models. Motivated by recent 2D video and multi-view image diffusion models, we aim to propose a diffusion-based method that generates multi-view videos of dynamic objects from text (see Fig. 1).

Compared to 2D video generation, T2MVid generation poses two new challenges. First, modeling multi-view videos is complex due to their four-dimensional nature, which involves different viewpoints as well as the dimensions of time and space (2D). Consequently, it is nontrivial for diffusion models to model such intricate data from scratch without extensive captioned multi-view video datasets. Second, there are no publicly available large-scale datasets of captioned multi-view videos, but it has been shown that billions of text and 2D image pairs are essential for powerful image diffusion models [74, 76, 83]. For example, Stable Diffusion [76] is trained on the massive LAION-5B dataset [78]. Unlike downloading 2D images available on the Internet, collecting a large quantity of multi-view videos is labor-intensive and time-consuming. This challenge is further compounded when high-quality captioned videos are needed, hindering the extension of diffusion models to T2MVid generation.

In this paper, instead of the labor-intensive task of collecting a large amount of captioned multi-view video data, we focus on the problem of enabling diffusion models to generate multi-view videos from text using only a comparable small dataset of captioned multi-view videos. This problem has not been taken into account by existing diffusion-based methods (*e.g.*, [6][23]). However, studies have revealed that naively fine-tuning a large pre-trained model on limited data can result in overfitting [30, 75, 115]. Our intuition is that we can factor the multi-view video generation problem into viewpoint-space and time components. The viewpoint-space component ensures that the generated multi-view videos are geometrically consistent and aligned with the input text, and the temporal component ensures temporal coherence. With such factorization, a straightforward approach is to leverage large-scale multi-view image datasets (*e.g.*, [51] [67]) and 2D video datasets (*e.g.*, Web10M [4]) to pre-train the viewpoint-space component and temporal component, respectively. However, while this approach can largely reduce the reliance on extensive captioned multi-view videos, it remains costly in terms of training resources.

Instead, we explore a new question: *can we jointly combine and reuse the layers of pre-trained 2D video and multi-view image diffusion models to establish a T2MVid diffusion model*? The large-scale pre-trained multi-view image diffusion models (*e.g.*, MVdream [80]) have learned how to model multi-view images, and the 2D temporal layers of powerful pre-trained video diffusion models (*e.g.*, AnimateDiff [23]) learned rich motion knowledge. However, new challenges are posed. We observe that naively combining the layers from these two kinds of diffusion models leads to poor generation

results. More specifically, the training data of multi-view image diffusion models are mainly rendered from synthetic 3D objects (*e.g.*, Objaverse [17, 16] ), while 2D video diffusion models are mainly trained on real-world 2D videos, posing a large domain gap issue.

To bridge this gap, we propose a novel diffusion-based pipeline, namely, Vivid-ZOO, for T2MVid generation. The proposed pipeline effectively connects the pre-trained multi-view image diffusion model [80] and 2D temporal layers[2] of the pre-trained video model by introducing two kinds of layers, named 3D-2D alignment layers and 2D-3D alignment layers, respectively. The 3D-2D alignment layers are designed to align features to the latent space of the pre-trained 2D temporal layers, and the introduced 2D-3D alignment layers project the features back. Furthermore, we construct a comparable small dataset consisting of 14,271 captioned multi-view videos to facilitate this and future research line. Although our dataset is much smaller compared to the billion-scale 2D image dataset (LAION [78]) and the million-scale 2D video dataset (e.g., WebVid10M [4]), our pipeline allows us to effectively train a large-scale T2MVid diffusion model using such limited data. Extensive experimental results demonstrate that our method effectively generates high-quality multi-view videos given various text prompts.

We summarize our contributions as follows:

- We present a novel diffusion-based pipeline that generates high-quality multi-view videos from text prompts. This is the first study on T2MVid diffusion models.

- We show how to combine and reuse the layers of the pre-trained 2D video and multi-view image diffusion models for a T2MVid diffusion model. The introduced 3D-2D alignment and 2D-3D alignment are simple yet effective, enabling our method to utilize layers from the two diffusion models across different domains, ensuring both temporal coherence and multi-view consistency.

- We contribute a multi-view video dataset that provides multi-view videos, text descriptions, and corresponding camera poses, which helps to advance the field of T2MVid generation.

## 2   Related work

**2D video diffusion model.** Many previous approaches have explored autoregressive transformers (*e.g.*, [18, 29, 107]), physical models [104] or GANs (*e.g.*, [8, 56, 77, 41]) for video generation. Recently, more and more efforts have been devoted to diffusion-based video generation [21, 26, 65, 94, 99, 102, 110, 121, 37], inspired by the impressive results of image diffusion models [11, 12, 74, 76, 83, 115].

The amount of available captioned 2D video data is significantly less than the vast number of 2D image-text pairs available on the Internet. Most methods [7, 22, 23, 90, 91, 92] extend pre-trained 2D image diffusion models to video generation to address the challenge of limited training data. Some methods employ pre-trained 2D image diffusion models (*e.g.*, [76]) to generate 2D video from texts in a zero-shot manner [36][118] or using few-shot tuning strategies [103]. These methods avoid the requirement of large-scale training data. Differently, another research line is to augment pre-trained 2D image diffusion models with various temporal modules or trainable parameters, showing impressive temporal coherence performance. For example, Ho et al. [28] extend the standard image diffusion architecture by inserting a temporal attention block. Animatediff[23] and AYL [7] freeze 2D image diffusion model and solely train additional motion modules on large-scale datasets of captioned 2D videos such as WebVid10M [4]. In addition, image-to-2D-video generation methods [6, 71, 105, 117] are proposed based on diffusion models to generate a monocular video from an image. Methods [95] focus on controllable video generation through different conditions such as pose and depth. MotionCtrl [98] and Direct-a-video [109] can generate videos conditioned by the camera and object motion. CameraCtrl [24] can also control the trajectory of a moving camera for generated videos. However, these text-to-2D-video diffusion models are designed for monocular video generation, which does not explicitly consider the spatial 3D consistency of multi-view videos.

**Multi-view image diffusion model.** Recent works have extended 2D image diffusion models for multi-view image generation. Zero123 [51] and Zero123++ [79] propose to fine-tune an image-conditioned diffusion model so as to generate a novel view from a single image. Inspired by this,

---

[2]For clarification, we add "2D" when referring to the layers of the 2D video diffusion models, while we add "3D" when referring to the multi-view image diffusion model.

many novel view synthesis methods [19, 32, 52, 53, 59, 93, 97, 100, 108, 112, 120] are proposed based on image diffusion models. For example, IM3D [59] and Free3D [120] generate multiple novel views simultaneously to improve spatial 3D consistency among different views. Differently, a few methods [13, 33, 89] adapt pre-trained video diffusion models (*e.g.*, [6]) to generate multi-view images from a single image. MVDream [80] presents a text-to-multi-view-image diffusion model to generate four views of an object each time given a text, while SPAD [34] generates geometrically consistent images for more views. Richdreamer [67] trains a diffusion model to generate depth, normal, and albedo.

**4D generation using diffusion models.** Many approaches [47, 51, 64, 69, 79, 80, 85, 96, 2] have exploited pre-trained diffusion models to train 3D representations for 3D object generation via score distillation sampling [64]. Recently, a few methods [3, 49, 70, 82, 113] leverage pre-trained diffusion models to train 4D representations for dynamic object generation. For example, Ling *et al.*[49] represent a 4D object as Gaussian spatting [35], while Bahmani *et al.*[3] adopt a NeRF-based representation [60, 61, 86]. Then, pre-trained 2D image, 2D video, and multi-view image diffusion models are employed to jointly train the 4D representations. In addition, diffusion models are used to generate 4D objects from monocular videos [14, 31]. Diffusion4D [46] presents a diffusion model that generates an orbital video around 4D content, and 4Diffusion [114] presents a video-conditioned diffusion model that generates MV videos from a monocular video. Different from all these methods, our approach focuses on presenting a T2MVid diffusion model. LMM [116] generates 3D motion for given 3D human models. DragAPart [43] can generate part-level motion for articulated objects. Unlike our method, Kuang *et al.*[38] focuses on generating multiple videos of the same scene given multiple camera trajectories.

# 3   Multi-view video diffusion model

**Problem definition.** Our goal for T2MVid generation is to generate a set of multi-view videos centered around a dynamic object from a text prompt. Motivated by the success of diffusion models in 2D video/image generation, we aim to design a T2MVid diffusion model. However, T2MVid generation is challenging due to the complexity of modeling multi-view videos and the difficulty of collecting massive captioned multi-view videos for training.

We address the above challenges by exploring two questions. (1) Can we design a diffusion model that effectively learns T2MVid generation, yet only needs a comparable small dataset of multi-view video data? (2) Can we jointly leverage, combine, and reuse the layers of pre-trained 2D video and multi-view image diffusion models to establish a T2MVid diffusion model? Addressing these questions can reduce the reliance on large-scale training data and decrease training costs. However, this question remains unexplored for diffusion-based T2MVid.

**Overview.** We address the above questions by factoring the T2MVid generation problem over viewpoint-space and time. With the factorization, we propose a diffusion-based pipeline for T2MVid generation (see Fig 2), including the multi-view spatial modules and multi-view temporal modules. Sec 3.1 describes how we adapt a pre-trained multi-view image diffusion model as the multi-view spatial modules. Multi-view temporal modules effectively leverage temporal layers of the pre-trained 2D video diffusion model with the newly introduced 3D-2D alignment layers and 2D-3D alignment layers (Sec 3.2). Finally, we describe training objectives in Sec 3.3 and the dataset construction to support our pipeline for T2MVid generation in Sec 3.4.

## 3.1   Multi-view spatial module

Our multi-view spatial modules ensure that the generated multi-view videos are geometrically consistent and aligned with the input text. Recent multi-view image diffusion models [34, 80] generate high-quality multi-view images by fine-tuning Stable Diffusion and modifying its self-attention layers. We adopt the architecture of Stable Diffusion for our multi-view spatial modules. Furthermore, we leverage a pre-trained multi-view image diffusion model based on Stable Diffusion by reusing its pre-trained weights in our spatial modules, which avoids training from scratch and reduces the training cost. However, the self-attention layers of Stable Diffusion are not designed for multi-view videos. We adapt these layers for multi-view self-attention as below.

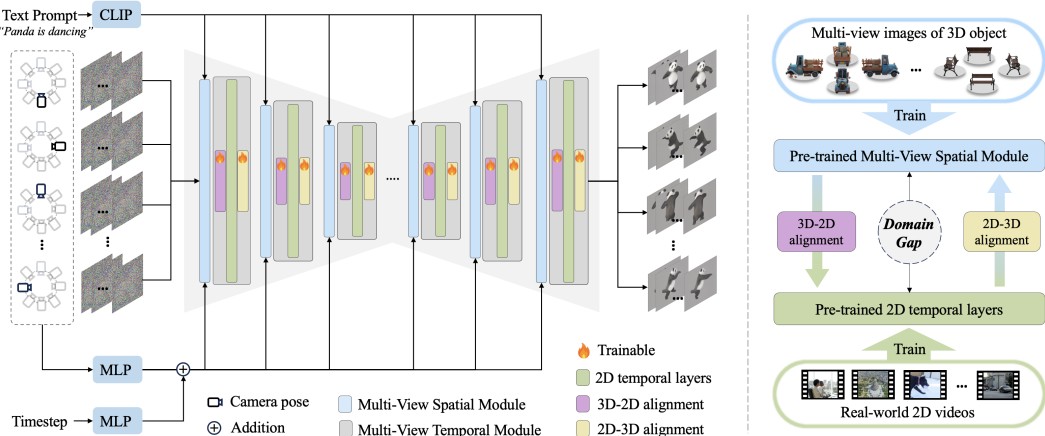

Figure 2: Overview of the proposed Vivid-ZOO. **Left**: Given a text prompt, our diffusion model generates multi-view videos. Instead of training from scratch, the multi-view spatial module reuses the pre-trained multi-view image diffusion model, and the multi-view temporal module leverages the 2D temporal layers of the pre-trained 2D video diffusion model to enforce temporal coherence. **Right**: Jointly reusing the pre-trained multi-view image diffusion model and temporal 2D layers poses new challenges due to the large gap between their training data (multi-view images of synthetic 3D objects versus real-world 2D videos). We introduce 3D-2D alignment and 2D-3D alignment to address the domain gap issue.

**Multi-view self-attention.** We inflate self-attention layers to capture geometric consistency among generated multi-view videos. Let $\mathbf{F} \in \mathbb{R}^{b \times K \times N \times d \times h \times w}$ denote the 6D feature tensor of multi-view videos in the diffusion model, where $b$, $K$, $N$, $d$ and $h \times w$ are batch size, view number, frame number, feature channel and spatial dimension, respectively. Inspired by [34, 80], we reshape $\mathbf{F}$ into a shape of $(b \times N) \times d \times (K \times h \times w)$, leading to a batch of feature maps $\tilde{\mathbf{F}}^n$ of 2D images, where $(b \times N)$ is the batch size, $\tilde{\mathbf{F}}^n$ denotes a feature map representing all views at frame index $n$, and $(K \times h \times w)$ is the spatial size. We then feed the reshaped feature maps $\tilde{\mathbf{F}}^n$ into self-attention layers. Since $\tilde{\mathbf{F}}^n$ consists of all views at frame index $n$, the self-attention layers learn to capture geometrical consistency among different views. We also inflate other layers of stable diffusion (see Appendix E) so that we can reuse their pre-trained weight.

**Camera pose embedding.** Our diffusion model is controllable by camera poses, achieved by incorporating a camera pose sequence as input. These poses are embedded by MLP layers and then added to the timestep embedding, following MVdream [80]. Here, our multi-view spatial module reuses the pre-trained multi-view image diffusion model MVDream [80].

## 3.2 Multi-view temporal module

Besides spatial 3D consistency, it is crucial for T2MVid diffusion models to maintain the temporal coherence of generated multi-view videos simultaneously. Improper temporal constraints would break the synchronization among different views and introduce geometric inconsistency. Moreover, training a complex temporal module from scratch typically requires a large amount of training data.

Instead, we propose to leverage the 2D temporal layers of large pre-trained 2D video diffusion models (*e.g.*, [23]) to ensure temporal coherence for T2MVid generation. These 2D temporal layers have learned rich motion priors, as they have been trained on millions of 2D videos (*e.g.*, [4]). Here, we employ the 2D temporal layers of AnimateDiff [23] due to its impressive performance in generating temporal coherent 2D videos.

However, we observed that naively combining the pre-trained 2D temporal layers with the multi-view spatial module leads to poor results. The incompatibility is due to the fact that the pre-trained 2D temporal layers and the multi-view spatial modules are trained on data from different domains (*i.e.*, real 2D and synthetic multi-view data) that have a large domain gap. To address the domain gap issue, one approach is to fine-tune all 2D temporal layers of a pre-trained 2D video diffusion model

on multi-view video data. However, such an approach not only needs to train many parameters but can also harm the learned motion knowledge [30] if a small training dataset is given. We present a multi-view temporal module (see Fig. 3) that reuses and freezes all 2D temporal layers to maintain the learned motion knowledge and introduce the 3D-2D alignment layer and the 2D-3D alignment layer.

**3D-2D alignment.** We introduce the 3D-2D alignment layers to effectively combine the pre-trained 2D temporal layers with the multi-view spatial module. Recently, a few methods [23, 6] add motion LoRA to 2D temporal attention for personalized/customized video generation tasks. However, our aim is different, *i.e.*, we expect to preserve the learned motion knowledge of 2D temporal layers, such that our multi-view temporal module can leverage the knowledge for ensuring temporal coherence.

Since motion prior knowledge is captured by the pre-trained 2D temporal attention layers, we insert the 3D-2D alignment layers before the 2D temporal attention layers. The 3D-2D alignment layers are learned to align the features into the latent space of the pre-trained 2D temporal layers. Furthermore, inspired by ControlNet [115] and [25], the 3D-2D alignment layers are inserted via residual connections and are zero-initialized, providing an identity mapping at the beginning of training. The process is described as follows:

$$\mathbf{F} = \alpha^{2D}(\mathbf{F}) + \alpha^{3D \rightharpoonup 2D}(\mathbf{F}) \tag{1}$$

where $\alpha^{3D \rightharpoonup 2D}$ is the 3D-2D alignment layer. $\alpha^{2D}$ is the 2D temporal layer followed by the 2D temporal attention layers and we refer to it as *2D in-layer* (see more details in Appendix). The 3D-2D alignment layer is plug-and-play and is simply implemented as an MLP.

**Multi-view temporal coherence.** We reuse and freeze the pre-trained 2D temporal layers in our multi-view temporal module to ensure the temporal coherence of each generated video. However, the 2D temporal layer is designed to handle 2D videos. We inflate the 2D temporal layer by reshaping the feature $\mathbf{F}$ to the 2D video dimension via the *rearrange* operation [73]. Then, 2D temporal layers $\gamma(\cdot)$ model temporal coherence across frames by calculating the attention of points at the same spatial location in $\mathbf{F}$ across frames for each video:

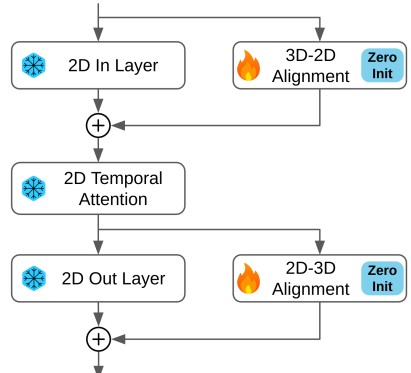

$$\mathbf{F} = \text{rearrange}(\mathbf{F}, \ b\,K\,N\,h\,w\,d \rightarrow (b\,K\,h\,w)\,N\,d) \tag{2}$$
$$\mathbf{F} = \gamma(\mathbf{F}) \tag{3}$$
$$\mathbf{F} = \text{rearrange}(\mathbf{F}, (b\,K\,h\,w)\,N\,d \rightarrow \ b\,K\,N\,h\,w\,d) \tag{4}$$

**2D-3D alignment.** We add the 2D-3D alignment layers after 2D temporal attention layers to project the feature back to the feature space of the multi-view spatial modules.

$$\mathbf{F}^a = \beta^{2D}(\mathbf{F}) + \beta^{2D \rightharpoonup 3D}(\mathbf{F}) \tag{5}$$

where $\beta^{3D \rightharpoonup 2D}$ is the 2D-3D alignment layer. $\beta^{2D}$ is the 2D temporal layer following the 2D temporal attention layer. The 2D-3D alignment layers are implemented as an MLP.

Figure 3: Our multi-view temporal module, where 3D-2D alignment layers are trained to align features to the latent space of the 2D temporal attention layers, and the 2D-3D alignment layers project them back.

### 3.3 Training objectives

We train our diffusion model to generate multi-view videos. Note that we freeze most layers/modules in the diffusion model and only train the 3D-2D and 2D-3D alignment layers during training, which largely reduces the training cost and reliance on large-scale data. Let $\mathcal{X}$ denote the training dataset, where a training sample $\{\mathbf{x}, y, \mathbf{c}\}$ consists of $N$ multi-view videos $\mathbf{x} = \{x\}_1^N$, $N$ corresponding camera poses $\mathbf{c}$, and a text prompt $y$. The training objective $\mathcal{L}$ on $\mathcal{X}$ is defined as follows:

$$\mathcal{L} = \mathbb{E}_{\mathbf{z}_t^v, y, \epsilon, t} \left[ \| \epsilon - \epsilon_\theta(\mathbf{z}_t^v, t, \tau_\theta(y), \mathbf{c}) \|^2 \right] \tag{6}$$

where $\tau_\theta(\cdot)$ is a text encoder that encodes the text into text embedding, $\epsilon_\theta(\cdot)$ is the denoising network. $\mathbf{z}_0^v$ is the latent code of a multi-view video sequence and $\mathbf{z}_t^v$ is its noisy code with added noise $\epsilon$.

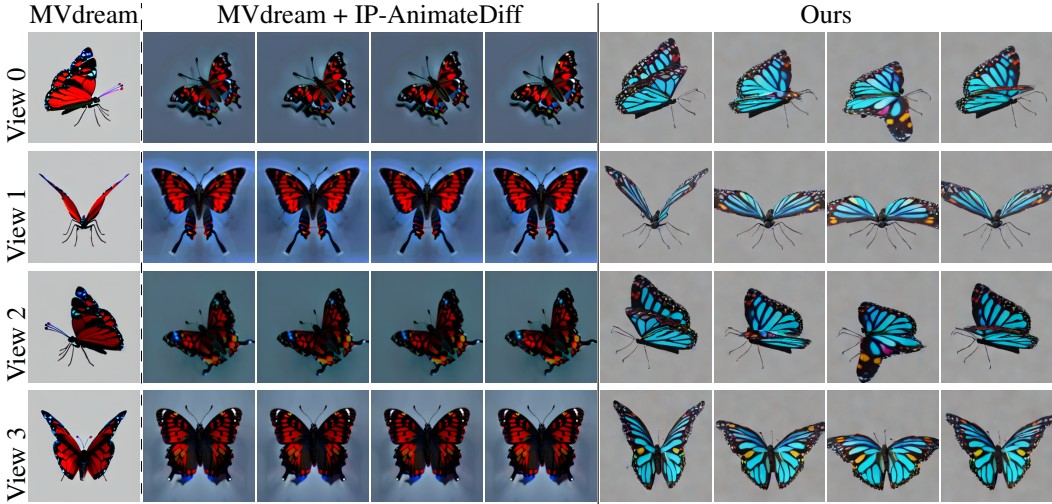

*Text prompt: Beautiful, intricate butterfly, 3d asset.*

Figure 4: Comparison on T2MVid generation. Although MVDream generates spatially 3D consistent images among views (the 1st column), MVDream + IP-AnimateDiff breaks the spatial 3D consistency among its generated videos. Instead, our method generates high-quality multi-view videos with large motions while maintaining temporal coherence and spatial 3D consistency.

### 3.4 Multi-view video dataset

Different from 2D images that are available in vast numbers on the Internet, it is much more difficult and expensive to collect a large amount of multi-view videos centered around 3D objects and corresponding text captions. Recently, multi-view image datasets (*e.g.*, [51, 67]), rendered from synthetic 3D models, have shown a significant impact on various tasks such as novel view synthesis [51, 93], 3D generation (Gaussian Splatting [84], large reconstruction model [50]), multi-view image generation [80] and associated applications. Motivated by this, we resort to rendering multi-view videos from synthetic 4D models (animated 3D models).

We construct a dataset named MV-VideoNet that provides 14,271 triples of a multi-view video sequence, its associated camera pose sequence, and a text description. In particular, we first select animated objects from Objaverse [17]. Objaverse is an open-source dataset that provides high-quality 3D objects and animated ones (*i.e.*, 4D object). We select 4D objects from the Objaverse dataset and discard those without motions or with imperceptible motions. Given each selected 4D object, we render 24-view videos from it, where the azimuth angles of camera poses are uniformly distributed. To improve the quality of our dataset, we manually filter multi-view videos with low-quality *e.g.*, distorted shapes or motions, very slow or rapid movement. For text descriptions, we adopt the captioning method Cap3D [57, 58] to caption a multi-view video sequence. Cap3D leverages BLIP2 [42] and GPT4 [63] to fuse information from multi-view images, generating text descriptions.

## 4 Experiments

**Implementation details.** We reuse the pre-trained MVDream V1.5 in our multi-view spatial module and reuse the pre-trained 2D temporal layers of AnimateDiff V2.0 in our multi-view temporal module. We train our model using AdamW [54] with a learning rate of $10^{-4}$. During training, we process the training data by randomly sampling 4 views that are orthogonal to each other from a multi-view video sequence, reducing the spatial resolution of videos to $256 \times 256$, and sample video frames with a stride of 3. Following AnimateDiff, we use a linear beta schedule with $\beta_{start} = 0.00085$ and $\beta_{end} = 0.012$. (Please refer to the Appendix for more details).

**Evaluation metrics.** Quantitatively evaluating multi-view consistency and temporal coherence remains an open problem for T2MVid generation. We quantitatively evaluate text alignment via CLIP [68] and temporal coherence via Frechet Video Distance (FVD) [87]. Yet, Ge *et al.*[20] pointed out FVD leans more towards per-frame quality than temporal consistency. To compensate for FVD,

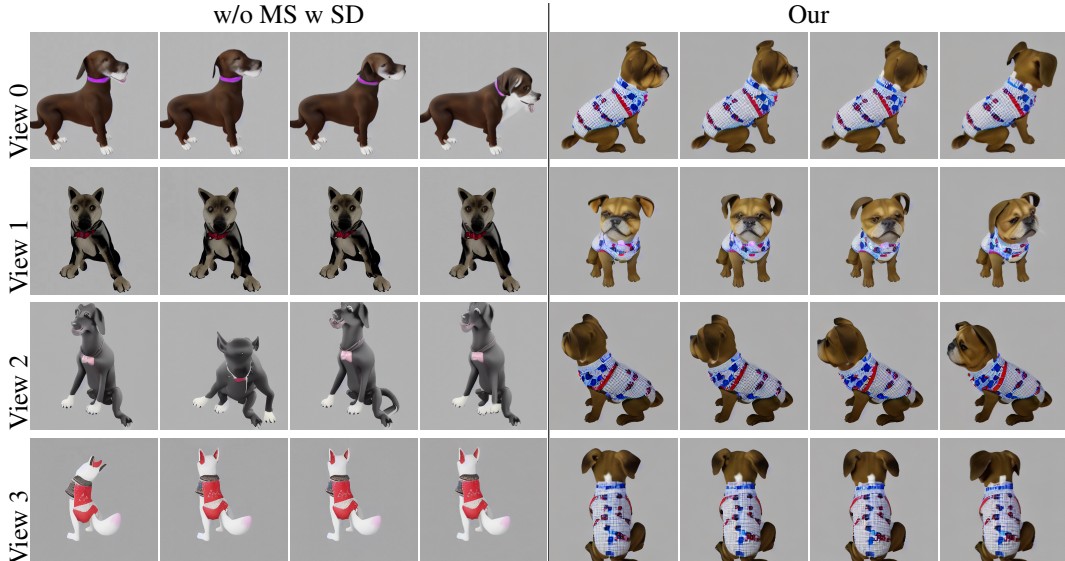

*Text prompt: a dog wearing an outfit, 3d asset*

Figure 5: Visual comparison of the contributions of our multi-view spatial module

we conduct a user study to evaluate the overall performance incorporating text alignment, temporal coherence, and multi-view consistency according to human preference (H. Pref.). CLIP and FVD scores in Tab. 1 are computed from 25 multi-view videos, where most input prompts used to generate these videos are separate from the training set, and only two prompts are from the training set. For ablation study, there are five methods and ten subjects for evaluating human preference, leading to 5×2×10 =100 questionnaires per input text prompt. To reduce the cost, we use input prompts to evaluate human preference in the ablation.

## 4.1 Qualitative and quantitative results

To the best of our knowledge, no studies have explored T2MVid diffusion models before. We establish a baseline method named **MVDream + IP-AnimateDiff** for comparison. **MVDream + IP-AnimateDiff** combines the pre-trained multi-view image diffusion model *MVDream* [80] and the 2D video diffusion model *AnimateDiff* [23], since MVDream generates high-quality multi-view images and AnimateDiff generates temporal coherent 2D videos. Following [119], we combine AnimateDiff with IP-adaptor [111] to enable AnimateDiff to take an image as input.

Given a text prompt, **MVDream + IP-AnimateDiff** generates multi-view videos in two stages, where MVDream generates multi-view images in the first stage, and IP-AnimateDiff animates each generated image from view into a 2D video in the second stage.

Fig. 4 and Tab. 1 show that **MVDream + IP-AnimateDiff** achieves slightly better CLIP values. However, our method outperforms **MVDream + IP-AnimateDiff** by a large margin in FVD and overall performance. **MVDream + IP-AnimateDiff** introduces the noticeable 3D inconsistency among different views. For example, both appearances and motions of the butterfly in the view 0 video are inconsistent with those of view 3. In contrast, our method not only achieves better performance in maintaining multi-view consistency, but also generates larger and more vivid motions for the butterfly, thanks to our pipeline and dataset. In addition, different from MVDream + IP-AnimateDiff employing two kinds of diffusion models and generating results in two stages, our method provides a unified diffusion model generating high-quality multi-view videos in only one stage. Please refer to the Appendix for more results.

## 4.2 Ablation study and discussions

We conduct the ablation study to show the effectiveness of the design in our multi-view spatial and temporal modules, as well as the proposed 3D-2D and 2D-3D alignment.

Table 1: Multi-view video generation. Best in bold.

| Method | FVD ↓ | CLIP ↑ | Overall ↑ |
|---|---|---|---|
| MVDream + IP-AnimateDiff | 2038.66 ± 44.36 | **32.71 ± 0.67** | 28% |
| Ours | **1634.28 ± 45.24** | 32.24 ± 0.78 | **72%** |

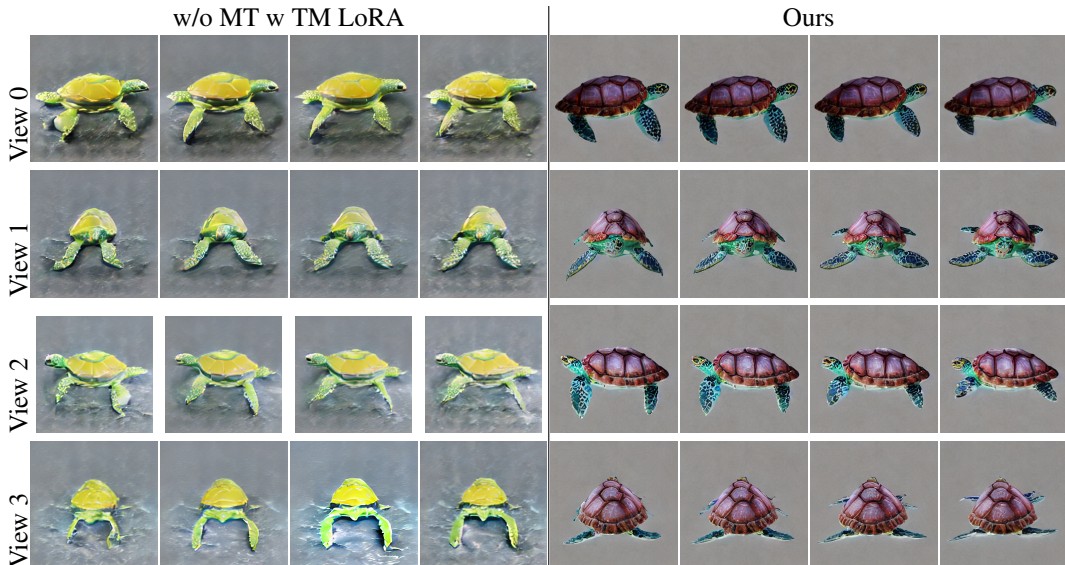

*Text prompt: a sea turtle, 3d asset.*

Figure 6: Visual comparison of the contributions of our multi-view temporal module

**Design of multi-view spatial module.** We build a baseline named **w/o MS w SD** that employs original Stable Diffusion 1.5 [76] as our multi-view spatial module and reuses its pre-trained weights. We also insert the camera embedding into the Stable Diffusion to enable viewpoint control. That is, **w/o MS w SD** is to generate a single-view video (2D) conditioned on input text and camera poses. We train **w/o MS w SD** on our dataset, where single-view videos are used as training data.

Since single-view video generation is much simpler than multi-view video generation, **w/o MS w SD** achieves high performance in video quality. However, **w/o MS w SD** fails to maintain multi-view consistency among different views (see Fig. 5) and has degraded overall generation performance (see Tab. 2). For example, the motion and shapes of the dragon are significantly inconsistent among views. Instead, by simply adapting a pre-trained multi-view image diffusion model as our spatial module, our method effectively ensures multi-view consistency.

**Design of multi-view temporal module.** Recent methods apply LoRA [30] to the 2D temporal attention layers of a pre-trained 2D video diffusion model and fine-tune only LoRA for personalized and customized 2D video generation tasks [23, 6, 72]. Following these methods, we build a temporal module named **TM LoRA** by inflating 2D temporal layers of AnimateDiff to handle multi-view videos and adding LoRA to the 2D temporal attention layers. We replace our multi-view temporal module with **TM LoRA**, and denote it by **w/o MT w TM LoRA**. Fig. 6 and Tab. 2 shows **w/o MT w TM LoRA** generates low-quality results,

Table 2: The ablation study results. The overall performance is assessed by a user study using paired comparison [3, 15].

| Method | Overall ↑ |
|---|---|
| w/o MS w SD | 44.88% |
| w/o MT w TM LoRA | 11.25% |
| w/o 3D-2D alignment | 53.50% |
| w/o 2D-3D alignment | 54.50% |
| Ours | **80.25%** |

despite being fine-tuned on our dataset. Instead, our multi-view temporal module inserts 3D-2D alignment and 2D-3D alignment layers before and after the 2D temporal attention layers, enabling the multi-view temporal module to be compatible with the multi-view spatial module.

**Effect of 3D-2D alignment.** We remove the proposed 3D-2D alignment from our model and train the model on our dataset with the same settings. Tab. 2 shows **w/o 3D-2D alignment** degrades

our temporal coherence and video quality performance. Instead, by projecting the feature to the latent space of the pre-trained 2D attention layers, our 3D-2D alignment layer effectively enables the 2D attention layers to align temporally correlated content, ensuring the video quality and temporal coherence.

**Effect of 2D-3D alignment.** As shown in Tab. 2, "w/o 2D-3D temporal alignment" achieves lower performance with the same training settings due to the removal of 2D-3D temporal alignment. The results indicate that only 3D-2D alignment is insufficient in jointly leveraging the pre-trained 2D temporal layers [23] and the multi-view image diffusion model [80] in our diffusion model. Instead, our 2D-3D alignment projects the features processed by the pre-trained 2D temporal layers back to the latent space of the multi-view image diffusion model, leading to high-quality results.

**Training cost.** MVDream is trained on 32 Nvidia Tesla A100 GPUs, which takes 3 days, and AnimateDiff takes around 5 days on 8 A100 GPUs. By combining and reusing the layers of MVDream and AnimateDiff, our method only needs to train the proposed 3D-2D alignment and 2D-3D layers, reducing the training cost to around 2 days with 8 A100 GPUs.

## 5  Conclusions

In this paper, we propose a novel diffusion-based pipeline named Vivid-ZOO that generates high-quality multi-view videos centered around a dynamic 3D object from text. The presented multi-view spatial module ensures the multi-view consistency of generated multi-view videos, while the multi-view temporal module effectively enforces temporal coherence. By introducing the proposed 3D-2D temporal alignment and 2D-3D temporal alignment layers, our pipeline effectively leverages the layers of the pre-trained multi-view image and 2D video diffusion models, reducing the training cost and accelerating the training of our diffusion model. We also construct a dataset of captioned multi-view videos, which facilitates future research in this emerging area.

## Acknowledgments and Disclosure of Funding

The research reported in this publication was supported by funding from King Abdullah University of Science and Technology (KAUST) - Center of Excellence for Generative AI, under award number 5940 and the SDAIA-KAUST Center of Excellence in Data Science and Artificial Intelligence. We thank Zhangjie Wu and Mengmeng Xu for their valuable constructive suggestions and help.

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

# Vivid-ZOO: Multi-View Video Generation with Diffusion Model
## — Supplementary Material —

In this appendix, we provide additional content to complement the main manuscript:

- Appendix A: Discussions about limitations of the current method and possible future improvements.
- Appendix B: Preliminaries about diffusion and latent diffusion models.
- Appendix C: Evaluation metrics for our multi-view video generation.
- Appendix D: Details about the multi-view captioned video dataset we construct.
- Appendix E: More details of our proposed model, including the spatial and temporal module.
- Appendix F: Additional qualitative visualized results.
- Appendix G: Societal impact, ethic concerns, dataset copyrights, and our safeguard policies.

## A  Limitations and future works

While our method takes a step forward in T2MVid generation, our method can be improved in a few aspects.

### A.1  Qualitative quality

For example, the visual quality of generated videos is not as high as that of multi-view image diffusion models due to the complexity of modeling multi-view videos. The spatial module of our method can be replaced with more advanced multi-view image diffusion models [34], to improve the performance of multi-view consistency. A large dataset of multi-view videos can be constructed, which further improves our method.

### A.2  Lighting

For lighting and rendering, we followed the settings of [80] to ensure fair comparisons. Since they used point light sources, our learned model also generates multi-view videos under the assumption of point light sources, which may result in different exposures across the viewpoints in the videos, as shown in Fig. III. Future work could explore generating videos that simulate more complex ambient lighting settings and even achieve controllable lighting for different viewpoints in the generation process.

### A.3  Topic of generation

Currently, the proposed Vivid-ZOO mainly supports generating multi-view videos for dynamic creatures with natural motions. Though we can also generate some categories like humans (*astronaut and horse*, Fig. II) and common objects (*waving flag*, Fig. IV), we believe our research can further inspire the community to develop more T2MVid techniques for the generation of more diverse and complex topics, similar to image diffusion model counterparts. For example, more specialized models that generate man-made artifacts like *moving cars* or *articulated furniture*, more powerful models that generate *multiple dynamic objects with complex motions*, and more diverse models like Vivid-Scene that generate dynamic scenes like *a stormy sea, an erupting volcano*.

## B  Background

**Diffusion model.** Diffusion models [27] learn to model a data distribution by iteratively recovering original data from noisy one, which comprises forward and backward phases. Given a clean sample $x_0$ from the data distribution $p_{data}$, the forward process gradually adds Gaussian noise to the sample, generating random latent variables $x_t$ at each time step $t \in [0, T]$:

$$q(x_t|x_{t-1}) = \mathcal{N}(x_t; \sqrt{1-\beta_t}x_{t-1}, \beta_t\mathbf{I}) \qquad (7)$$

where $\beta_t$ is a hyperparameter that determines the noise schedule. With a large time step, $x_T$ is assumed to be perturbed into a standard Gaussian noise. Given $x_T$, the denoising network is trained to gradually remove the noise and recover the original data:

$$p_\theta(x_{t-1}|x_t) = \mathcal{N}(x_{t-1}; \mu_\theta(x_t, t), \Sigma_\theta(x_t, t)) \tag{8}$$

where $\theta$ denote parameters of the denoising network, $\mu$ and $\Sigma$ are mean and variance, respectively.

**Latent diffusion model** (LDM). By embedding images into low-dimensional latent codes, latent diffusion models [74] (LDMs) perform the diffusion process in the latent space of latent codes, significantly reducing the computational cost. Typically, LDMs employ a pre-trained autoencoder (*e.g.*, VQ-VAE [88]), which consists of an encoder and decoder, where the encoder transforms images into the latent space and the decoder maps denoised latent codes back to the pixel space.

## C   Details about evaluation metrics

- *Multi-view Text alignment*: Text-to-2D-video diffusion models (*e.g.*, [23, 36, 102]) adopt CLIP score [68] to measure the alignment between an input text and a corresponding generated 2D video. To evaluate the alignment between the input text and generated multi-view videos, we first measure the CLIP score for each view and then average the CLIP scores for all views.

- *Video quality*: We adopt Frechet Video Distance (FVD) to measure the quality of generated multi-view videos. FVD is the standard metric adopted by many 2D video generation [95, 106] and animation methods [6, 105]. FVD measures the quality of generated videos by measuring the data distribution between generated and training videos, where I3D networks pre-trained on the Kinetics dataset [10] are employed to extract features.

- *Human preference.* We conduct a user study to measure the overall quality of generated multi-view video text alignment, temporal coherence and multi-view consistency.

  We adopt paired comparison. We invite 10 subjects to participate in the user study. For each subject, we display two multi-view video sequences generated by different generation methods as well as the corresponding input text prompt, where the two results are arranged in an up-and-down order, and a resulting multi-view video sequence is displayed in a single row. We told each subject that the task is to generate four orthogonal views of a dynamic 3D object. Then the subject was asked to choose a multi-view video sequence whose overall quality is better in text alignment, temporal coherence and multi-view consistency. For comparison with MVdream + IP-AnimateDiff, we use 10 input prompts to generate multi-view videos for the user study. For the ablation study, there are five methods in total, leading to more combinations of paired comparisons. We hence use 5 input prompts in the use study for the ablation study.

## D   More details about our multi-view video dataset

2D video diffusion models [6] have pointed out that data curation is essential to improve the generation performance of diffusion models. The training of 2D Video diffusion models can be degraded if the training dataset contains many static 2D videos [6]. Hence, we developed an animated 3D object selection tool that automatically discards 4D objects that are static or close to static. In particular, given a 4D object, we first render a video from a single viewpoint. For efficiency, instead of using advanced optical flow algorithms, we calculate the pixel difference between different frames in each video, in order to identify whether 4D objects are static.

With the selected 4D objects, we employ Cycles [3] as the rendering engine to render multi-view videos. Given a 4D object, we render 24-view videos with the resolution of $512 \times 512$. We first uniformly distribute the camera poses around the normalized 4D object and then add subtle disturbances. The radius of a camera to a 4D object is in the range of [2.2 2.6], and a camera's height is in [0.8 1.2]. The background of a multi-view video sequence is randomly filled in gray color. The frame number of multi-view video sequences is diverse, depending on its 4D objects.

To further improve the quality of our dataset, we manually discard low-quality data from our dataset. We found that many multi-view videos contain distorted shapes or motions. We remove these low-quality data to avoid their negative effect on the training generation models. On the other hand, we

---

[3]https://www.cycles-renderer.org/

also remove multi-view videos that contain large translation motions. Due to the large translation motions, objects disappear in a few frames, which leads to these frames having only a background. In addition, many 4D objects are textureless in Objaverse [17]. We keep 10% of these textureless objects. For caption generation, we adopt a caption method *i.e.*, Cap3D which is designed to caption multi-view images of a 3D object. The Cap3D is used to describe a multi-view frame sequence sampled from a multi-view video sequence.

# E    More implementation details

**Training settings.** Table II provides detailed information on the hyperparameter settings and hardware configuration used for model training. During training, four orthogonal views are randomly chosen, leading to four-view videos. For a video from a viewpoint, the starting frame is randomly selected, and then we extract one frame every 3 frames. The frame size is $256 \times 256$ and the frame number is set to 16 (see Table I).

Table I: Training dataset settings

| Name | Parameter value |
| --- | --- |
| view number | 4 |
| sample size | $256 \times 256$ |
| sample stride | 3 |
| frame number | 16 |

Table II: Training settings

| Name | Parameter value |
| --- | --- |
| noise scheduler type | DDIMScheduler |
| noise scheduler timesteps number | 1000 |
| noise scheduler start beta | 0.00085 |
| noise scheduler end beta | 0.012 |
| noise scheduler beta schedule | linear |
| noise scheduler steps offset | 1 |
| noise scheduler clip sample | false |
| optimizer | AdamW |
| learning rate | 0.0001 |
| train step number | 100000 |
| batch size | 16 |
| CPU memory size in total | 320G |
| GPU type | NVIDIA A100 |
| GPU number | 8 |

**Inference settings.** Table III lists the hyperparameters and hardware configurations in the inference stage. The resolution and number of frames in the generated video are the same as those in the training settings.

Table III: Inference settings

| Name | Parameter value |
| --- | --- |
| sample step number | 50 |
| CFG weight | 7.5 |
| CPU memory | 30G |
| GPU type | NVIDIA A100 |
| GPU number | 1 |

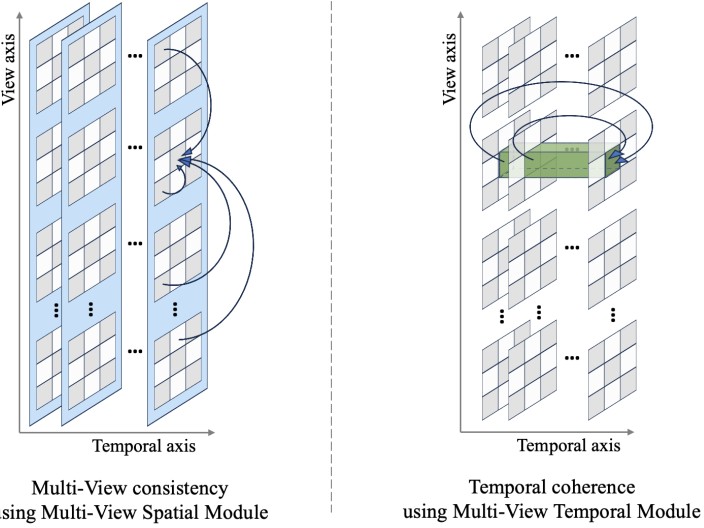

Figure I: The multi-view spatial module of our method ensures multi-view consistency of generated multi-view videos via capturing correlations across different views. The multi-view temporal module enforces temporal coherence via capturing temporal correlations among frames in a video of a viewpoint.

### E.1    More details about multi-view spatial module

Our multi-view spatial module adapts Stable Diffusion to handle multi-view videos, and reuses the pre-trained weight of MVDream[80]. In this main paper, we have elaborated how to adapt Stable Diffusion's self-attention layers to handle multi-view videos' 6D feature tensors. With the adaption, the self-atention layers model multi-view consistency among views, as shown in Fig. I. For other layers, we first reshape the features of multi-view videos using the rearrange operation: rearrange($\mathbf{F}$, $b\ K\ N\ h\ w\ d \rightarrow (b\ N)K\ h\ w\ d$). When the features are fed to the multi-view temporal module, we transform the dimensions of the output feature $\mathbf{F}'$ back with rearrange operation: ($\mathbf{F}'$, $(b\ N)K\ h\ w\ d \rightarrow b\ K\ N\ h\ w\ d$).

### E.2    More details of multi-view temporal module

Fig. I shows how our multi-view temporal module leverages 2D temporal layers to caption temporal correlations among frames in each view video.

Both a **3D-2D Alignment** layer and **2D-3D Alignment** layer are implemented using a Linear MLP layer. We experimented with 2-layer/3-layer setups, but there was no improvement in performance. Therefore, we use a simple single layer for implementation. In this paper, we reuse AnimateDiff in the multi-view temporal module, where the 2D-in-layer refers to the "project_in" layer and 2D-out-layer refers to "_out" layer in the AnimateDiff.

## F    Additional results

### F.1    Additional text to multi-view video examples

Some additional experimental results are presented from Fig. II to Fig. VIII. We strongly recommend the readers to watch the corresponding videos on our anonymous website to get a better feel for the movement of the objects in the picture.

### F.2    More ablation study results

Fig. IX and Fig. X show the contributions of the 3D-2D and 2D-3D alignment layers respectively.

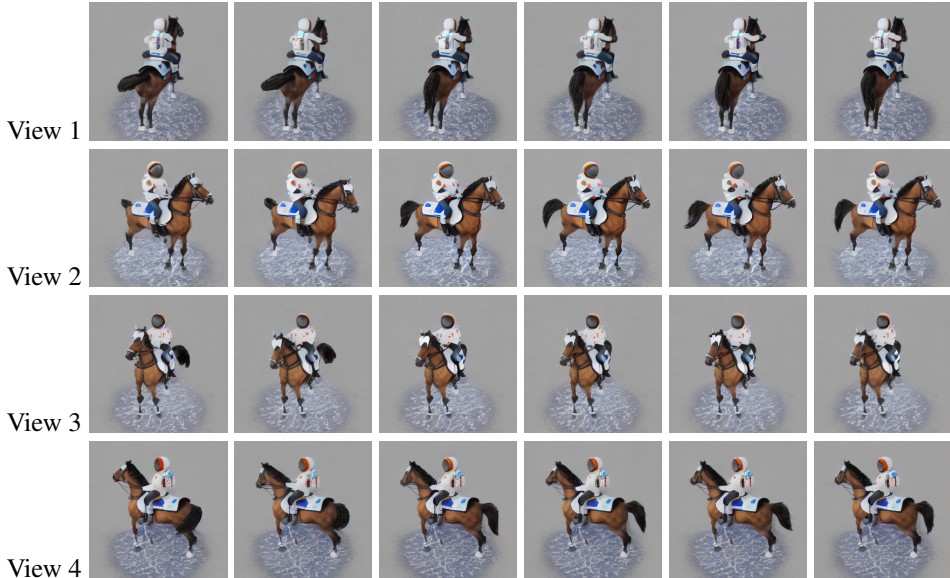

Figure II: Text prompt: *an astronaut riding a horse, 3d asset*

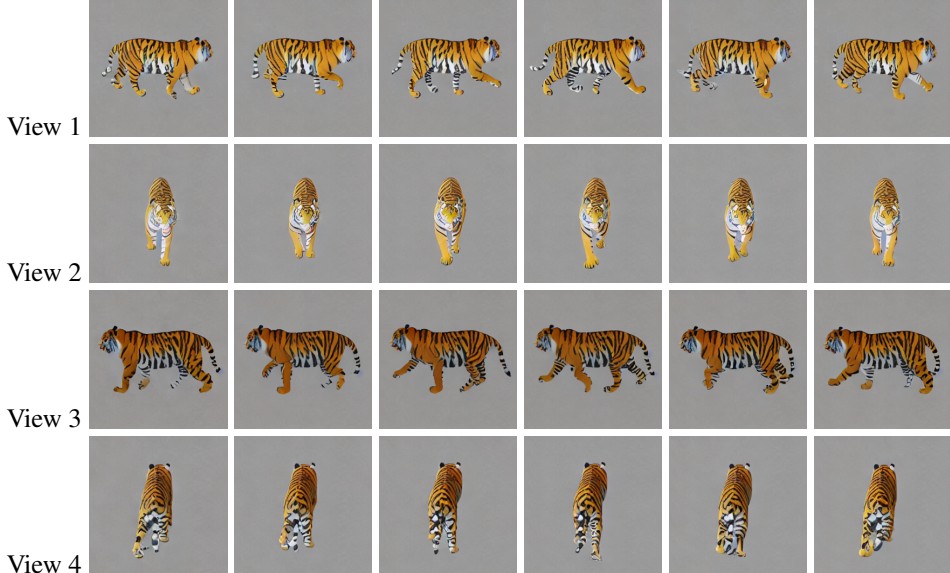

Figure III: Text prompt: *A full-bodied tiger walking, 3d asset*

# G  Societal impact and ethic concerns

## G.1  Positive societal impact

Our method is able to generate vivid multi-view videos for dynamic creatures. Therefore, our method can be directly applied to enhance creativity and entertainment, e.g., creating AR/VR and game assets. Due to the availability of multi-view videos, our method can also be used for art creation and interactive educational content, which could benefit many people, like artists, designers, educators, and film and television creators. Researchers in fields such as biology, ecology, and zoology can benefit from this technology by creating accurate multi-view visualizations of dynamic creatures, which can aid in research, analysis, and presentations.

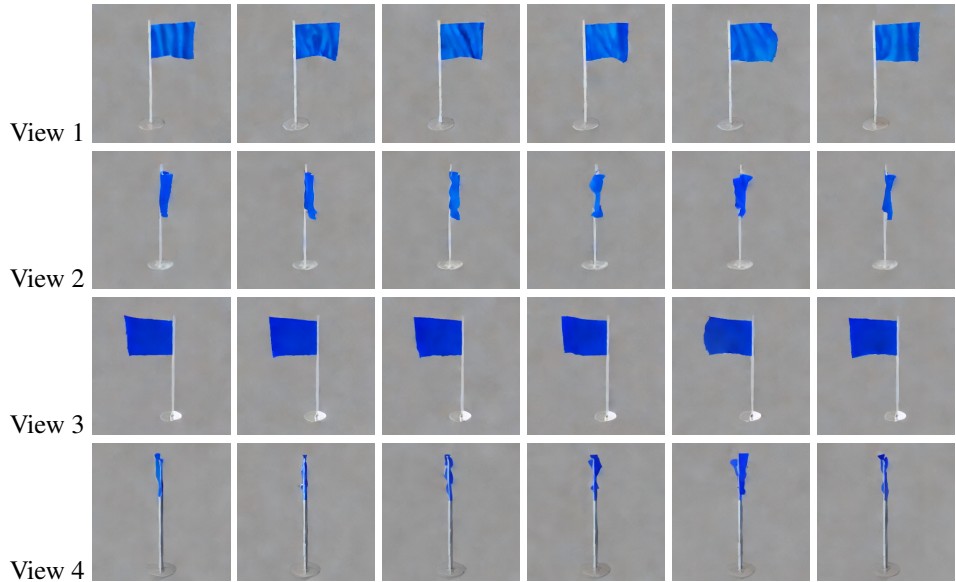

View 1

View 2

View 3

View 4

Figure IV: Text prompt: *a blue flag attached to a flagpole, with a smooth curve, 3d asset*

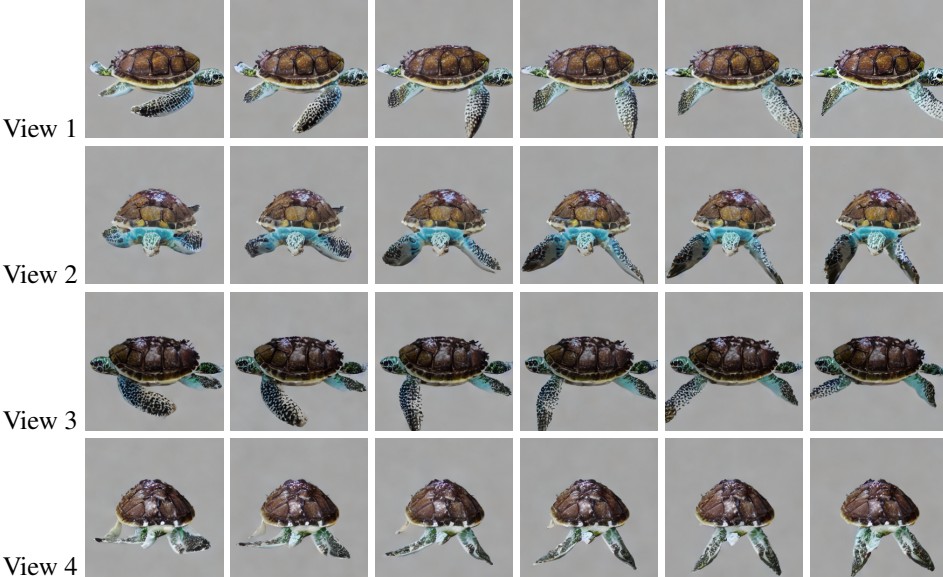

View 1

View 2

View 3

View 4

Figure V: Text prompt: *a spiked sea turtle, 3d asset*

## G.2 Negative societal impact

Our Text-to-Multi-view-Video diffusion method is based on one existing text-to-multi-view image model and one text-to-video model. Therefore, its internal representation may inherit some bias from these two base models. Our multi-view video generation could be exploited to create highly realistic deepfakes. These fake videos can be used for malicious purposes such as spreading disinformation, manipulating public opinion, or creating fake profiles for fraudulent activities. If users input provocative prompts or maliciously fine-tune the model parameters, our model could potentially generate harmful videos, such as those containing vulgarity, gore, or violence. However, since our model is fine-tuned on the dynamic creature dataset, we believe the risk of such content is significantly lower compared to previous open-domain generative models like MV-Dream [80] and SVD [6]. Additionally, we will implement gated access and usage guidelines for our model and continuously monitor community usage and feedback to prevent such harmful content as much as possible.

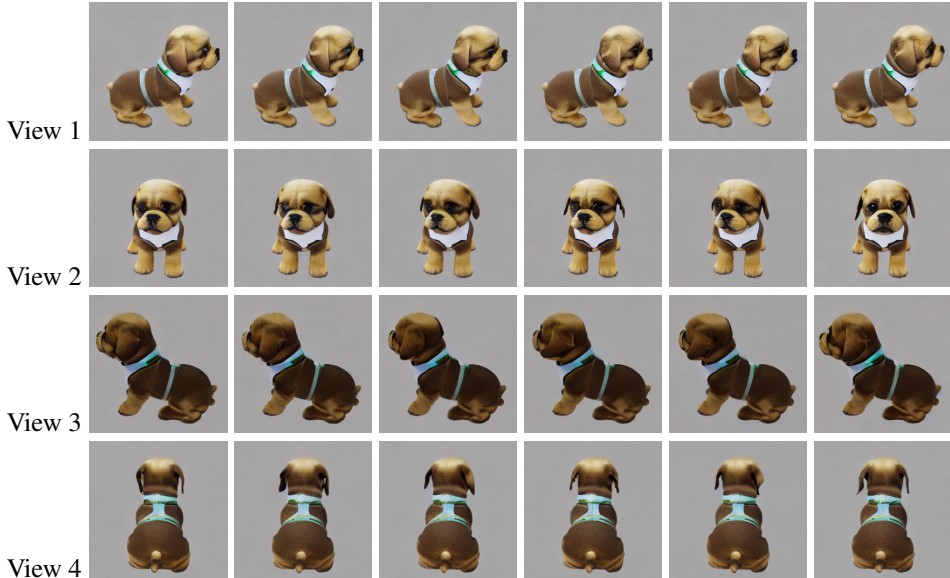

View 1

View 2

View 3

View 4

Figure VI: Text prompt: *a dog wearing a outfit, 3d asset*

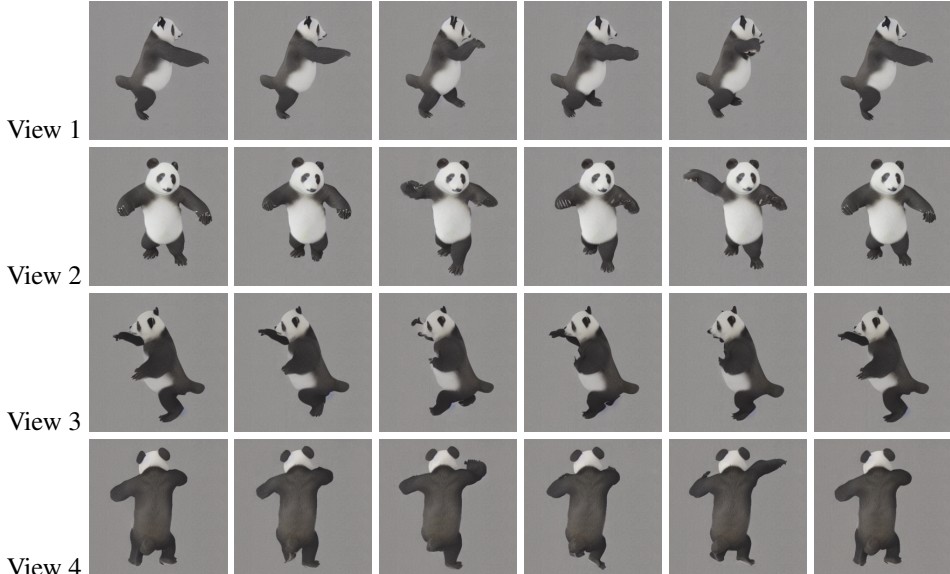

View 1

View 2

View 3

View 4

Figure VII: Text prompt: *a panda is dancing*

## G.3 Copyright

Our dynamic dataset is directly sourced from the already open-sourced and published Objaverse [17] dataset and is used solely for scientific research purposes. Therefore, it does not infringe on the legal rights and copyrights of the original 3D/4D model creators and collectors.

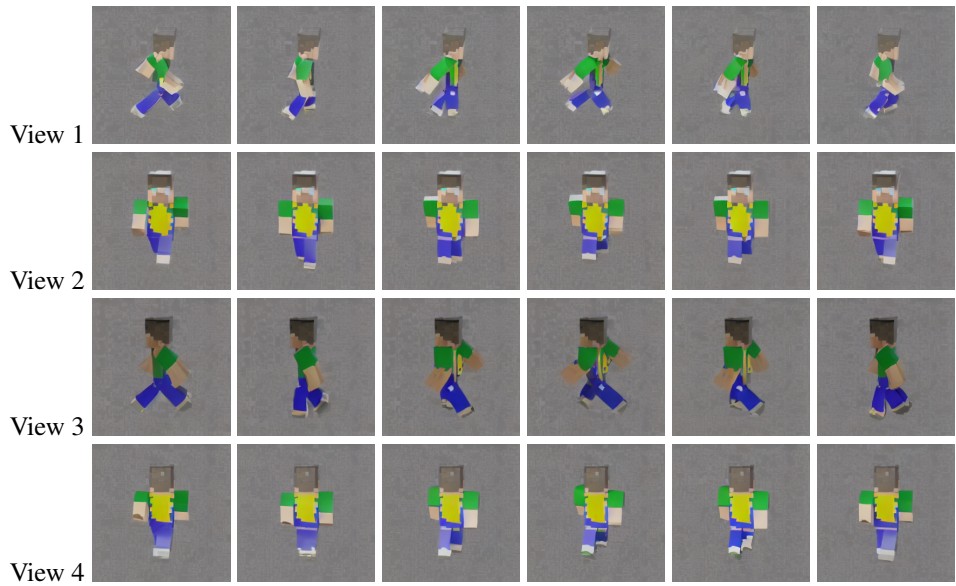

Figure VIII: Text prompt: *a pixelated Minecraft character walking, 3d asset*

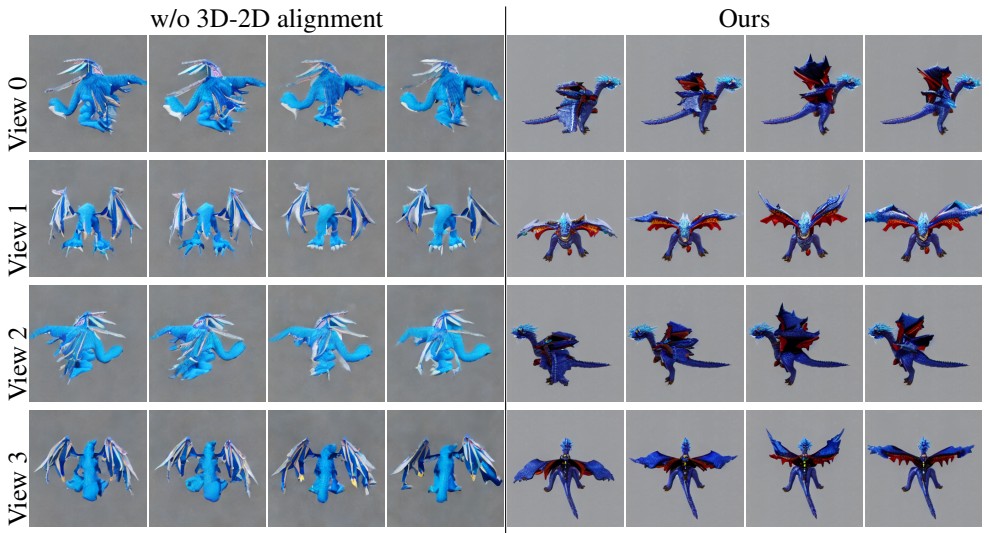

*Text prompt: a blue-winged dragon, also depicted as a flying monster, 3d asset*

Figure IX: Visual comparison of the contributions of our 3D-2D alignment layers

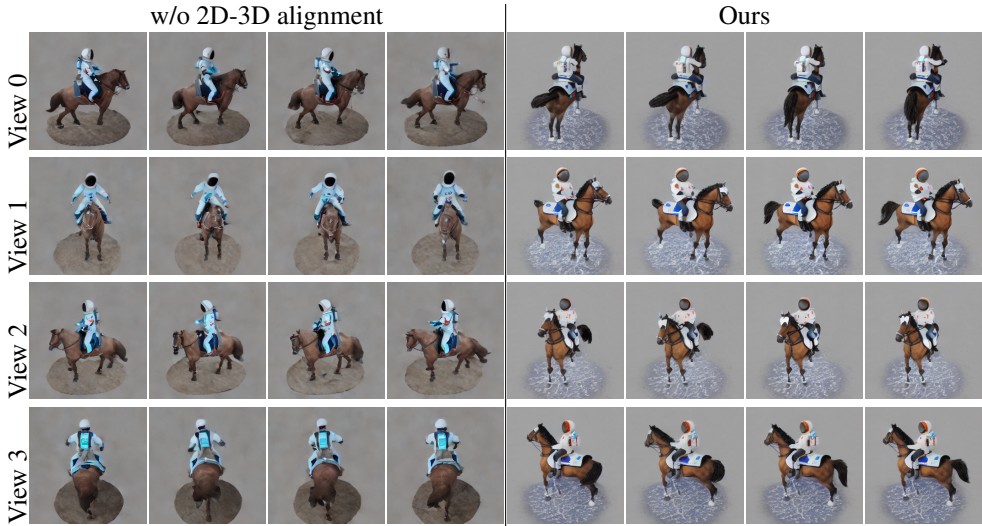

*Text prompt: an astronaut riding a horse, 3d asset*

Figure X: Visual comparison of the contributions of our 2D-3D alignment layers

