# OpenReview forum: "Vivid-ZOO: Multi-View Video Generation with Diffusion Model"
_NeurIPS.cc/2024/Conference — NeurIPS 2024 poster_

### Official Review · Reviewer_VRVx · 2024-07-08

**Soundness:** 3
**Presentation:** 3
**Contribution:** 3
**Rating:** 7
**Confidence:** 2

**Summary:**

The paper introduces "Vivid-ZOO," an innovative diffusion model designed for Text-to-Multi-view-Video (T2MVid) generation. This is a novel approach that addresses the challenges of generating high-quality multi-view videos from textual descriptions. Specifically, the authors propose a new method that leverages diffusion models to generate multi-view videos centered around dynamic 3D objects from text prompts. The T2MVid generation problem is effectively factorized into viewpoint-space and time components, allowing for the combination and reuse of layers from advanced pre-trained multi-view image and 2D video diffusion models. And the authors claim that it is the first study that explores the application of diffusion models to the T2MVid generation task.

**Strengths:**

1. The paper introduces a novel diffusion-based approach to generate multi-view videos from text prompts, which is a relatively unexplored area in the literature.The creative factorization of the problem into viewpoint-space and time components is an innovative way to tackle the complexity of multi-view video generation.The introduction of alignment modules to integrate pre-trained models is a clever solution to address domain gaps and reuse of layers.
2. Furthermore, by providing a captioned multi-view video dataset, the authors not only support their own research but also contribute to the wider research community, facilitating future work and lowering the entry barriers for other researchers in this domain.
3. The proposed solution adeptly addresses a notable gap in the current literature by offering an efficient approach that does not rely on massive datasets, thereby making sophisticated multi-view video generation more accessible and applicable to a wider array of problems

**Weaknesses:**

1. The visual quality of the generated videos might not match state-of-the-art single-view image generation models. Future work could focus on improving the visual fidelity and realism of the generated multi-view videos.
2. The model currently assumes point light sources, which might not always produce the most realistic renderings. Enhancements in simulating complex lighting conditions could improve the model's applicability.

**Questions:**

Please refer to weakness

**Limitations:**

The authors discuss the limitations and potential negative societal impacts of their work in the paper.

---

> ### Author Rebuttal · Authors · 2024-08-07
>
> We appreciate the reviewer for the positive comments and constructive suggestions.  We are encouraged that the reviewer agrees that  "an innovative diffusion model",  "novel approach", "a clever solution to address domain gaps", "contribute to the wider research community”, and "offering an efficient approach".
>
>
> **W1: Future work could focus on improving the visual fidelity and realism of the generated multi-view videos.**
>
> Thank you for your constructive suggestions. In our future work, we will jointly train our model on real-world 2D videos and our dataset, leveraging real-world 2D videos to improve the visual fidelity and realism of the generated multi-view videos.
>
> **W2: Enhancements in simulating complex lighting conditions could improve the model's applicability.**
>
> Thank you for the suggestion. We haven't considered lighting conditions. We will explore complex lighting conditions to improve the model's applicability. Inspired by your suggestion, we will also improve our dataset by exploring complex lighting conditions. We plan to explore more advanced lighting models to better simulate complex lighting scenarios, such that a higher-quality and more realistic dataset can further improve the generation performance of the model.

---

> > ### Comment · Area_Chair_ZH1b · 2024-08-12
> > **Please discuss**
> >
> > Dear reviewer,
> >
> > The discussion period is coming to a close soon. Please do your best to engage with the authors.
> >
> > Thank you,
> > Your AC

---

> ### Author Response · Authors · 2024-08-13
>
> Dear Reviewer VRVx,
>
> We sincerely thank the reviewer for your positive feedback.  Your insightful comments help us further improve the paper's quality.
>
> Best,
>
> The Authors.

---

### Official Review · Reviewer_CcuB · 2024-07-09

**Soundness:** 3
**Presentation:** 3
**Contribution:** 3
**Rating:** 5
**Confidence:** 5

**Summary:**

This paper proposes a Text-to-Multi-view-Video generation algorithm capable of generating multi-view video content based on text descriptions. The method decouples the multi-view spatial and temporal dimensions, utilizing pretrained models like MVDream for multi-view generation and animatediff for video generation. It addresses domain gaps between different features through 3D-2D/2D-3D alignment. The authors trained on captioned 4D data from objverse and conducted various ablation studies to validate their approach.

**Strengths:**

1. The method proposed by the authors achieved impressive visual results, with performance metrics showing improvements over naively using MVDream + AnimateDiff.
2. The 3D-2D/2D-3D alignment proposed by the authors reduces the number of parameters and training time needed. Given the limited training data, this approach also helps mitigate degradation in pretrained models.

**Weaknesses:**

1The authors claim their method is the first to do text-to-multi-view video generation, but as far as I know, there are many works related to 4D generation currently, such as MAV3D, AYG, 4D-fy, including recent works like 4Diffusion and Diffusion4D. I believe the authors may want to emphasize that their work focuses on novel view synthesis rather than involving 4D representation, such as Deformable GS, Dynamic Nerf, etc. However, these are just differences at the algorithmic level; fundamentally, they are all engaged in multi-view video generation. Therefore, the authors' comparison in Table 1 is insufficient, and relevant 4D generation works should also be included for comparison.

**Questions:**

1. Regarding the 'overall' metric in Table 1, does it refer to the results of a user study, or is it a comprehensive representation of the previous two results?
2. What is the test dataset in Table 1? How large is the dataset?
3. Will the 4D dataset proposed in the paper be made publicly available?
4. In the current framework, what are the results if MVDream and AnimateDiff are not fixed, or if fine-tuning is done with a very small learning rate?
5. Did the authors conduct experiments on generating 4D contents based on image prompt, and are there any visual results?

**Limitations:**

Currently, the results only compare against a self-designed baseline, lacking a broader comparison with other algorithms.

---

> ### Author Rebuttal · Authors · 2024-08-07
>
> We appreciate the reviewer for the positive comments and constructive suggestions. We are encouraged that the reviewer agrees that our work "achieved impressive visual results", "performance metrics showing improvements", "reduces the number of parameters and training time needed", and "mitigate degradation in pretrained models."
>
>
> **W1.1. The authors claim their method is the first to do text-to-multi-view video generation.**
>
> Thank you for your comments. Our claim is that our method is ``the first study on T2MVid **(Text-to-Multi-view-Video) diffusion models**", instead of the first to do text-to-multi-view video generation,  as stated in line 82 of the main paper.
>
> Different from MAV3D, AYG and 4D-fy which use the pre-trained diffusion models to optimize 4D representations, we focus on presenting a diffusion model that generates Multi-View (MV) videos from texts.
> The concurrent work Diffusion4D presents a diffusion model that generates **an orbital video** around 4D content, and 4Diffusion presents a **video-conditioned** diffusion model that generates MV videos from a monocular video. The focus of these two methods is different from ours. We will cite the Diffusion4D and 4Diffusion.
> Since the concurrent Diffusion4D and 4Diffusion appeared after our submission, we will provide detailed discussions in our final version.
>
>
>
>
> **W1.2. Including relevant 4D generation works for comparison.**
>
> Thank you for your constructive suggestions. Following your suggestions, we treat 4D-fy, AYG [43], and MAV3D [76] as MV video generation methods and then compare our method with these methods.
>
> Since AYG [43] and MAV3D [76] have not released their code, we obtained their generated videos from their websites and conducted qualitative experiments. As shown in Figs. A and B of the global response, our method outperforms both MAV3D and AYG. AYG employs motion amplification to enhance its generated motions, yet, this distorts the magic wand (see Fig.A in the global response).
> The table below shows our method achieves better performance than 4D-fy on ten testing samples.
>
> | Method | FVD ↓ | Overall↑ |
> |--------|-------|----------|
> | 4D-fy  | 2189.87 | 37%     |
> | Ours   | **1621.92** | **63%** |
>
>
>
> **Q1. 'overall' metric in Table 1.**
>
> The overall metric refers to the results of a user study using paired comparison. The evaluation metric in lines 264-265 of the main paper provides more details.
> We will clarify the overall metric in Table 1 in our final version.
>
>
>
> **Q2. Testing dataset in Table 1.**
>
> To the best of our knowledge, there is no established and widely used testing dataset for testing text-to-multi-view video generation.
> Furthermore, the testing data in existing text-to-4D generation methods (e.g., 4D-fy, MAV3D, AYG) are different from each other.
>
> In Table 1, we used **25** diverse text prompts (i.e., testing dataset) to evaluate our method, where 11 prompts are shown on the web mentioned in the abstract. Noting that some text prompts are taken from existing 3D or 4D generation methods to challenge our method, for example, the prompt used in Fig. 4 in the main paper comes from MVDream, the prompt in Fig. II comes from MVDream and 4D-fy [3], and the prompt in Fig. VII comes from AYG [43] and MAV3D [76]. We will release our testing dataset in our final version.
>
>
> **Q3. Will the 4D dataset proposed in the paper be made publicly available?**
>
> Yes. Our 4D dataset will be made publicly available.
>
> **Q4. What are the results if MVDream and AnimateDiff are not fixed.**
>
> We only re-use the temporal layers of AnimateDiff instead of all its layers. When both MVDream and the temporal layers of AnimateDiff are not fixed, the GPU memory required for model training is substantially increased, since the number of trainable parameters is significantly increased. Training such a model, **runs out of memory** on 8 A100 (80G) using the same setting as our method. Similarly, when MVDream is not fixed, the model also **runs out of memory** with the same setting.
>
> Additionally, we build another baseline, namely, Ours (trainable T-AnimateDiff), that does not fix the temporal layers of AnimateDiff in our method.  The trainable parameters of this baseline increase by 600\%,   significantly escalating the difficulties and computational cost of training the diffusion model. The table below shows the performance of our method is largely degraded, when the temporal layers of AnimateDiff are not fixed.
>
> | Method                       | FVD ↓   |
> |------------------------------|---------|
> | Ours (trainable T-AnimateDiff) | 2796.86 |
> | Ours                          | **1634.28**  |
>
>
> **Q5. Did the authors conduct experiments on generating 4D content based on image prompt**
>
> We haven't conducted experiments based on image prompts. In this paper, we mainly focus on the text-to-multi-view-video diffusion model. Thank you for your suggestions.   Image-to-4D is another interesting and meaningful topic, we will explore it in our future work.

---

> > ### Comment · Reviewer_CcuB · 2024-08-12
> >
> > Thank you for the thorough response. The rebuttal has addressed most of my concerns. I'd like to keep my initial score

---

> ### Author Response · Authors · 2024-08-13
>
> Dear Reviewer CcuB,
>
> We sincerely thank the reviewer for your positive feedback. Your constructive suggestions help us improve the paper's quality.
>
> Best,
>
> The Authors.

---

### Official Review · Reviewer_K81K · 2024-07-10

**Soundness:** 2
**Presentation:** 3
**Contribution:** 3
**Rating:** 5
**Confidence:** 4

**Summary:**

This paper focuses on generating multi-view consistent videos given a text prompt, specifically from 4 orthogonal views. It fine-tunes MVDream with a temporal layer adopted from AnimateDiff. To mitigate the domain gap between the two layers, some connected layers called 3D-2D and 2D-3D alignment layers are introduced. They train the newly introduced layers on a 4D subset of Objaverse. The proposed model can generate multi-view videos. The authors conduct evaluations to verify the new layers and choices of the base models.

**Strengths:**

* This is the first paper to study the problem of text-conditioned multi-view video generation, based on my best knowledge. Perhaps [1] is a concurrent work and should be acknowledged.
* The paper is easy to follow.
* A multiview video dataset extracted from Objaverse is collected.

[1] Kuang, Zhengfei, et al. "Collaborative Video Diffusion: Consistent Multi-video Generation with Camera Control." arXiv preprint arXiv:2405.17414 (2024).

**Weaknesses:**

Despite the fact that the paper is the first to study text-conditioned multi-view video generation, I think there could still be room for improvement in the evaluation section in terms of the justification of the method as stated below:

1. There are very few generated videos (8 in the supplementary material) shown in the paper. Is this because the success rate is low? Since it is a forward-based method, I expect the time to generate the videos will not be too long. If so, it would be good to discuss the failure cases in the limitation section.

2. As for human evaluation, only five videos are used for comparison.

3. There is a lack of details about how many videos are used to compute CLIP and FVD scores. Also, are the prompts used to generate these videos separate from the training set? FVD was used to evaluate temporal coherence, but FVD is shown to be not good at it (see [2]).

    [2] Ge, Songwei, et al. "On the Content Bias in Fréchet Video Distance." CVPR. 2024.

4. The motions in most generated videos are small and are not indicated even in the text prompt. This could limit the use case of the model in generating specific motions.

5. All the generated videos contain a grayish background. I'm fine with it if the ultimate goal is to generate a 4D asset, and I think it is a reasonable goal. Ideally, like MVDream, the authors may want to compare with 4D generation methods.

**Questions:**

1. As far as I know, MVDream is fine-tuned (fully?) from SD2.1, while AnimateDiff v2 is fine-tuned from SD1.5; this could cause an additional discrepancy when combining the two models, right?
2. In the ablation study of "Design of multi-view spatial module", what is the difference between what you do and directly using AnimateDiff v2?

**Limitations:**

The authors discuss the limitations and negative societal impact in the appendix.

---

> ### Author Rebuttal · Authors · 2024-08-07
>
> We would like to thank the reviewer for the constructive comments. We are encouraged that the reviewer agrees that our work is "the first paper to study",  "easy to follow", and "A multiview video dataset ... is collected".
>
> **S1: This is the first paper to study the problem of text-conditioned multi-view video generation.**
>
> We appreciate your recognition of our contributions. Unlike our method, concurrent work [Kuang et al.] focuses on generating multiple videos of the same scene given multiple camera trajectories. We will cite [Kuang et al.].
>
>
>
> **W1: Success rate of the proposed method.**
>
> The success rate of our method is not low. We used diverse prompts to challenge our method, including some from existing methods: the prompt used in Fig. 4 in the main paper is from MVDream, the prompt in Fig. II is from MVDream and 4D-fy [3], and the prompt in Fig. VII is from AYG [43] and MAV3D [76].
>
> The 25 Multi-View (MV) videos generated by our method (11 videos on the website mentioned in the abstract) achieve better performance in Tab. 1 of the main paper, showcasing the effectiveness of our method.  We provide more results in the PDF of the global response, as we cannot upload videos due to the NeurIPS 2024 regulations.
>
>
> **W2: As for human evaluation, only five videos are used for comparison.**
>
> As stated in line 783 in Appendix C, we used **ten** sequences of MV videos (text prompts), for **comparison**.
> As to the ablation study, there are five methods and ten subjects, leading to 5×2×10 =100 questionnaires per MV video sequence. To reduce the cost, we followed 4D-fy [3] and used five MV videos in the ablation.
> As suggested, we increase the video number to 10 in the table below, showing our method outperforms the baselines in the ablation.
>
> | Method                | Human evaluation↑ |
> |--|--|
> | w/o MS w SD           | 44.88%            |
> | w/o MT w TM LoRA      | 11.25%            |
> | w/o 3D-2D alignment   | 53.50%            |
> | w/o 2D-3D alignment   | 54.50%            |
> | Ours                  | **80.25%**            |
>
>
>
> **W3.1: Number of MV videos to compute the CLIP and FVD scores.**
>
> Thank you for your valuable suggestion. The number is 25. We will clarify this in our final version.
>
>
> **W3.2: Are prompts separate from the training set?**
>
> Yes. Most prompts used to generate these videos are separate from the training set, and only two prompts are from the training set. We will clarify this in our final version.
>
>
> **W3.3: FVD is shown to be not good at evaluating temporal coherence [R2].**
>
> we adopt FVD, since many 2D video generation methods (e.g. [6, 89, 98, 99]) use it to evaluate temporal coherence. To compensate for FVD, we also used human evaluation in Tab.1.  Following the suggested paper [Ge et al.], we calculate FVD-new by replacing I3D with videoMAE. The table below shows that our method consistently achieves better performance on 25 samples.
>
> | Method                  | FVD-new [Ge et al.] ↓ |
> |--|--|
> | MVDream + IP-AnimateDiff| 466.82           |
> | Ours                    | **459.31**                   |
>
>
> **W4: The motions in most generated videos are small and are not indicated even in the text prompt.**
>
> The motions in at least **five** generated videos are not small (see the web mentioned in the abstract and Figs. 4, III,  VII, VIII, IX). And the motions in the **five** generated videos are indicated in text prompts (see Figs. II, III,  VII, VIII, IX).
> We intentionally omit motion descriptions in some prompts to evaluate how our method performs with various text prompts. Despite such omission, our method still generates natural and vivid motions for the corresponding objects rather than static or improper ones.
> We provide more results that contain motion descriptions in Fig. C (see PDF of global response).
>
>
> It is worth noting that our method re-uses the pre-trained temporal layers of AnimateDiff in our temporal module, instead of training from scratch, **"minimizing the requirement for large-scale data and extensive computational resources"**, as recognized by reviewer MoTs. Despite such reductions, we find that the **motion magnitude** of generation results by **our method is much larger** than MVdream + IP-AnimateDiff (see Fig. 4), thanks to our dataset and our 3D-2D and 2D-3D alignment layers.
> Our motion generation performance can be further improved by replacing AnimateDiff with a 2D video diffusion model with better motion capture performance.
>
>
> **W5: Comparison with 4D generation methods.**
>
> Due to the time limitation,  according to constructive suggestions raised by Reviewer CcuB, we compare our method with 4D generation methods 4D-fy, AYG [43], and MAV3D [76] by treating them as MV video generation methods.
> Since AYG [43] and MAV3D [76] have not released their code, we take generated videos from their website to conduct qualitative experiments. As shown in Figs. A and B in the PDF of global response, our method outperforms both MAV3D and AYG.  AYG uses motion amplification to amplify its generated motions; however, it distorts the magic wand (see Fig.A in the global response).
>
>
> The table below shows that our method outperforms 4D-fy on ten samples. Since 4D-fy needs around 18 hours to train a 4D object, due to the time limitation, we obtain only ten 4D-fy results.
>
> | Method | FVD-new [Ge et al.] ↓ | Overall↑ |
> |--|--|--|
> | 4D-fy  | 661.57               | 37%      |
> | Ours   | **443.99**           | **63%**  |
>
>
>
> **Q1: MVDream is fully fine-tuned from SD2.1.**
>
> MVDream had indeed provided a model fine-tuned from SD 1.5 on its GitHub page. Hence, there is no additional discrepancy.
>
> **Q2: In the ablation study of "Design of multi-view spatial module", what is the difference between what you do and directly using AnimateDiff v2.**
>
>
> The largest difference is that AnimateDiff v2 does not have camera embedding, which can not enable viewpoint control. Moreover, we finetune the baseline on our dataset, while  AnimateDiff v2 is trained on Web10M.

---

> > ### Comment · Reviewer_K81K · 2024-08-11
> > **Response**
> >
> > Thank you for your time and effort in addressing the unclear details, adding additional experiments, and performing additional evaluations. I have got most of my concerns addressed and updated my rating.

---

> ### Author Response · Authors · 2024-08-13
>
> Dear Reviewer K81K,
>
> We sincerely thank the reviewer for your positive feedback. Your constructive review comments help us improve the quality of the manuscript.
>
> Best regards,
>
> The Authors.

---

### Official Review · Reviewer_MoTs · 2024-07-13

**Soundness:** 3
**Presentation:** 3
**Contribution:** 3
**Rating:** 7
**Confidence:** 4

**Summary:**

The paper presents a novel diffusion-based pipeline designed to generate multi-view videos from text prompts. The core challenge addressed by the paper is the generation of dynamic 3D object videos from multiple viewpoints, a task not extensively explored with existing diffusion models. The authors factor the problem into viewpoint-space and time components, allowing for the reuse of layers from pre-trained multi-view image and 2D video diffusion models. This reuse is intended to ensure consistency across views and temporal coherence, thereby reducing training costs. The paper also introduces 3D-2D and 2D-3D alignment layers to bridge the domain gap between the training data of multi-view images and 2D videos. Additionally, a new dataset of captioned multi-view videos was created.

**Strengths:**

1. The paper tackles a relatively unexplored area of diffusion models, extending their application to multi-view video generation from text prompts. The task is extremely challenging and important.
2. By reusing layers from existing models, the approach leverages the strengths of pre-trained systems while minimizing the requirement for large-scale data and extensive computational resources, addressing a common barrier in training new machine learning models.
3. The introduction of 3D-2D and 2D-3D alignment layers is an interesting solution to the issue of domain gaps between different types of training data.
4. The creation of a new, although smaller, dataset for training and testing in this niche area is crucial, as it provides a resource that can be built upon by subsequent research. However, this does not seem to be license-free?

**Weaknesses:**

1. The paper acknowledges the challenge of not having a large-scale, captioned multi-view video dataset, which is crucial for training robust diffusion models. The authors attempt to mitigate this by constructing a smaller dataset, but this could limit the model’s generalization capabilities and performance across diverse scenarios. Further, this may be very challenging if we would like to further scale to scene-level contents.
2. The alignment between pre-trained 2D video and multi-view image diffusion models is crucial, yet challenging. The proposed 3D-2D and 2D-3D alignment layers are a solution, but the effectiveness of these layers in truly bridging the domain gap might not fully compensate for the inherent differences in training data types (real-world 2D videos versus synthetic multi-view data), potentially affecting the quality of generated videos. Since the model is trained on an entirely different domain of object-centric videos, there is very little we can tell in terms of video quality degradation.

**Questions:**

N/A

**Limitations:**

Yes

---

> ### Author Rebuttal · Authors · 2024-08-07
>
> We appreciate the reviewer for the positive comments and constructive suggestions. We are encouraged that the reviewer agrees that "The task is extremely challenging and important", "tackles a relatively unexplored area of diffusion models,", "a novel diffusion-based pipeline", "a new dataset",  "minimizing the requirement for large-scale data and extensive computational resources", and "an interesting solution".
>
> **S4. License of the new dataset.**
>
> We greatly appreciate your recognition of our contributions. Our dataset is built from 4D assets provided by Objaverse [15], which are released under the ODC-BY license. This license allows us to make our dataset available using the same license.
>
> **W1: The authors attempt to mitigate the challenge of lacking a large-scale dataset by constructing a smaller dataset, but this could limit the model’s generalization capabilities and performance across diverse scenarios.**
>
> Thank you for your valuable comments. Yes, ideally, if we could construct a large-scale dataset of millions or billions of Multi-View (MV) video sequences, similar to 2D image/video diffusion methods, the method could achieve excellent generalization.
> However, the cost of constructing such a large-scale dataset is enormous.
>
>
> Instead, although we create a smaller dataset, we resort to re-using the layers of the pre-trained MV image and 2D video diffusion model which have learned rich knowledge from large-scale data of MV images and 2D videos, to improve the generalization performance of our method. We observe that our method can generate meaningful results that are rather irrelevant to our MV video dataset, since our method effectively leverages the knowledge learned by pre-trained MV image and 2D video diffusion models.  Our method opens up a new path, though its generalization performance is not yet optimal.
>
>
> We appreciate your suggestions. In our future work, we will further improve our dataset, including its size and diversity. For scene-level contents, we can similarly construct a small dataset of scene-based MV videos to train our method while replacing MVDream with a pre-trained scene-based MV image diffusion model.
>
>
>
> **W2: The alignment between pre-trained 2D video and multi-view image diffusion models is crucial, yet challenging.**
>
> Thank you for your comments. We agree that it is challenging to bridge the domain gap between real-world 2D videos and synthetic multi-view data, due to the inherent differences. To reduce the difficulty in bridging the domain gap,
> our 3D-2D alignment is to project the feature into the latent space of the pre-trained 2D video diffusion model's 2D temporal attention layers, instead of all temporal layers. In other words, the temporal attention layers capture temporal correlation among features, which can reduce the difficulty of alignment to some extent.
> Consequently, although a 3D-2D alignment layer is implemented as only an MLP, our method can generate high-quality MV videos.
>
> We are inspired by your valuable comments. To further enhance the quality of the generated videos, we will jointly train our model on real-world 2D videos, and our dataset, such that real-world 2D video data is leveraged to preserve the original video generation performance.

---

> > ### Comment · Area_Chair_ZH1b · 2024-08-12
> > **Please discuss**
> >
> > Dear reviewer,
> >
> > The discussion period is coming to a close soon. Please do your best to engage with the authors.
> >
> > Thank you,
> > Your AC

---

> ### Author Response · Authors · 2024-08-13
>
> Dear Reviewer MoTs,
>
> We sincerely thank the reviewer for your positive feedback. Your constructive review comments help us enhance the strength of our work.
>
> Best regards,
>
> The Authors.

---

### Author Rebuttal · Authors · 2024-08-07

We thank all reviewers for their constructive comments and valuable suggestions. We are encouraged by the reviewers' positive feedback on novelty, writing, methodology, and  experiment, such as

- novelty: "a novel approach", (VRVx), "a novel diffusion-based pipeline" (MoTs),  "an innovative diffusion model" (VRVx), "an interesting solution" (MoTs), and "the first paper to study" (K81K).

- writing: "easy to follow" (K81K).

- methodology: "minimizing the requirement for large-scale data and extensive computational resources" (MoTs),     "reduces the number of parameters and training time needed" (CcuB), and "a clever solution to address domain gaps" (VRVx).

- experiment: "achieved impressive visual results" (CcuB), and "performance metrics showing improvements" (CcuB).




**To Reviewer K81K.**

**Q1: More multi-view videos generated by our method.**
Following your suggestion, we provide additional multi-view generation results of our method in Fig. C of the attached PDF. Moreover, motion descriptions are included in the text prompts. Fig. C shows our method achieves high generation performance in terms of large motion magnitude and text alignment.


**To Reviewers K81K and CcuB.**

**Q1: Compared with 4D generation method.**
Due to the time limitation, following constructive suggestions raised by Reviewer CcuB, we compare our method with 4D generation methods 4D-fy, AYG [43] and MAV3D [76] by treating them as multi-view video generation methods.  As shown in Figs. A and B in the attached PDF, While AYG employs motion amplification to enhance its generated motions, this approach results in distortion of the magic wand, as shown in Fig. A of the global response. The table below shows that our method outperforms 4D-fy on ten samples.
| Method | FVD ↓ | Overall↑ |
|--------|-------|----------|
| 4D-fy  | 2189.87 | 37%     |
| Ours   | **1621.92** | **63%** |

---

### Decision · Program_Chairs · 2024-09-25

**Decision:**

Accept (poster)

**Comment:**

The paper received two Accepts and two Borderline Accepts. The reviewers appreciated the new task, the proposed dataset, and some of the results and ideas. However, there are still concerns about the comparisons with existing about generalizations / performance at scale and comparisons. I agree with all these points mentioned by the reviewers and recommend Accept as poster.